# Studies on the Removal of Congo Red Dye by an Adsorbent Based on Fly-Ash@Fe₃O₄ Mixture

**Maria Harja** [1], **Nicoleta Lupu** [2], **Horia Chiriac** [2], **Dumitru-Daniel Herea** [2] and **Gabriela Buema** [2,*]

1  Faculty of Chemical Engineering and Environmental Protection "Cristofor Simionescu", "Gheorghe Asachi" Technical University of Iasi, 700050 Iasi, Romania
2  National Institute of R&D for Technical Physics, 700050 Iasi, Romania
*  Correspondence: gbuema@phys-iasi.ro

**Abstract:** The effectiveness of a $Fe_3O_4$-loaded fly ash composite for the adsorption of Congo red dye was assessed in this work. The structure and properties of the magnetic adsorbent were established by scanning electron microscopy (SEM), energy dispersive spectroscopy (EDS), X-ray diffractometer (XRD), vibrating sample magnetometer (VSM), and dynamic light scattering (DLS). The magnetic results showed a saturation magnetization value of 6.51 emu/g and superparamagnetic behavior. The main parameters that influence the removal of Congo red dye adsorbent such as dose, initial concentration, and contact time were examined. The Freundlich adsorption isotherm and pseudo-second-order kinetic model provided the best fit for the experimental findings. The Congo red dye's maximum adsorption capacity of 154 mg/g was reported in the concentration range of 10–100 mg/L, using the proposed magnetic adsorbent. The results of the recyclability investigation demonstrated that the circular economy idea is valid. The adsorbent that was synthesized was also further characterized by XRD and FTIR techniques after Congo red dye adsorption.

**Keywords:** fly ash; NaOH; $Fe_3O_4$; Congo red adsorption; equilibrium isotherms; kinetics models; circular economy

## 1. Introduction

The presence of Congo red dye in water is not acceptable because it prevents water from re-oxygenating. Exposure to this pollutant leads to eye and skin irritation, stomach pain, nausea, vomiting, and diarrhea [1]. The adsorption process has been utilized as a suitable model for Congo red removal. Thus, various adsorbents with different adsorption capacities have been proposed [2–11].

Fly ash is the byproduct of burning coal in thermal power plants. Although it has typically been used as a starting material in different fields (building materials, geopolymers, catalysts, stabilizers for land in certain areas), there has also been a specific focus on the use of fly ash as an adsorbent due to its availability in high quantities as an industrial waste product [12]. Additionally, fly ash has a high $Al_2O_3$ and Si content, which highlights its potential as an adsorbent [13].

Fly ash can be used as-cast, or modified to remove contaminants from aqueous solutions. The literature has reported that modified fly ashes are more effective in the adsorption process because they have a larger surface area than unmodified fly ashes [14]. Fly ash can be converted into an inexpensive and high-performance adsorbent by simple modifications due to its unique porous properties [15]. The alkalis used for the modification include sodium/potassium hydroxide, sodium carbonate etc. [16], while the acids that can be used are hydrochloric acid, sulfuric acid, nitric acid, oxalic acid, citric acid and acetic acid [17]. Adsorption studies using fly ash treated with NaOH solution have attracted attention and are numerous. The characterization by various basic techniques has revealed significant changes in the fly ash structure. In addition, the specific surface areas of synthesized

materials are higher compared to unmodified fly ash. As-prepared materials based on NaOH-treated fly ash represent a way to solve the problems associated with its storage and to obtain clean and safe wastewater.

Iron oxides are mineral compounds with several crystal structures as well as various structural and magnetic characteristics [18]. In nature, there are different types of iron oxides, e.g., hematite ($\alpha$-$Fe_2O_3$), magnetite ($Fe_3O_4$), and maghemite ($\gamma$-$Fe_2O_3$). The physical properties of these compounds, including their lattice parameter (nm), crystallographic structure, magnetic status, density, color, and hardness can be found in the published literature [19].

Magnetite ($Fe_3O_4$) also known as loadstone, black iron oxide, or ferrous ferrite exhibits the strongest magnetization of any transition metal oxide [20] and represents one of the most interesting crystallographic phases of iron oxide [18]. The methods used in the laboratory to synthesize $Fe_3O_4$ include the electrochemical method [21,22], co-precipitation method [23–26], sol–gel method [27], sol–gel explosion-assisted method [28], hydrothermal chemical method [29], and the mechanochemical hydrothermal approach [20,30].

Magnetic nanoparticles must have several particular characteristics in order to be involved in the treatment of water contaminated with various pollutants, such as a strong magnetic response, rapid reuse, chemical stability, and low toxicity [31,32]. Unfortunately, there are some drawbacks in the utilization of magnetic nanoparticles including their low adsorption capacity, sensitivity to oxidation, and tendency to agglomerate [33].

In our previous papers, it was demonstrated that unmodified fly ash [34] has the potential to remove Congo red from an aqueous solution with a maximum adsorption capacity of 22.12 mg/g. The capacity of $Fe_3O_4$ to remove Congo red dye was also reported in [35]. However, the adsorption capacity needs to be increased. Sivalingam and Sen prepared nanozeolite X synthesized by the hybrid ultrasonic method from fly ash and used the adsorbents for Congo red removal. The results of their study showed that the materials that were prepared had adsorption capacities in the range of 88.64 mg/g–110.24 mg/g [36].

According to our previous research, fly ash treated with NaOH with different working parameters of synthesis (contact time, method of synthesis, fly ash/NaOH ratio, NaOH concentration) [37] and fly ash treated with $Fe_3O_4$ [38] can be used as adsorbents for wastewater contaminated with various pollutants.

Therefore, the purpose of this study was to develop a material based on the treatment of unmodified fly ash with NaOH and $Fe_3O_4$ for the removal of Congo red dye. The first step consists of the direct activation method of fly ash with NaOH, 10 M, at room temperature (RT). Obtaining magnetic adsorbent is the outcome of the second step. Specific material characterization via scanning electron microscopy (SEM), energy dispersive spectroscopy (EDS), X-ray diffractometer (XRD), vibrating sample magnetometer (VSM), and dynamic light scattering (DLS) methods was carried out. The adsorption behavior of Congo red dye using the magnetic adsorbent was studied with a focus on the effects of the magnetic adsorbent dosage, initial Congo red dye concentration, and contact time. The data were fitted using Langmuir and Freundlich isotherm models in their linear and nonlinear forms. The pseudo-first-order model, pseudo-second-order model, and intraparticle diffusion model were employed for the kinetic study. The concept of a circular economy is also presented based on the recyclability test. The XRD and FTIR results for the loaded adsorbent are presented at the end of the study.

To the best of our knowledge, there are no available studies based on the treatment of fly ash with NaOH and $Fe_3O_4$ and its subsequent use as an adsorbent for Congo red dye removal from an aqueous solution. Moreover, this type of mixture represents an innovative category of magnetic materials with excellent adsorption capacities. This fact is underlined by the comparison between the material synthesized in this study, Fly-ash@$Fe_3O_4$, and other magnetic adsorbents described in the literature.

## 2. Materials and Methods

All reagents were used without further purification. NaOH was purchased from Chemical Company (Iasi, Romania). Congo Red dye, FeCl$_2$·4H$_2$O and FeCl$_3$·6H$_2$O were obtained from Merck KGaA, Darmstadt, Germany.

### 2.1. Fly-Ash/NaOH Adsorbent Synthesis

Fly ash was collected from a power plant in Iasi (CET II Holboca, Romania). A direct activation technique was used to modify fly ash adsorbent. First, the fly ash was mixed with 10 M NaOH at room temperature (RT) for 4 h [39]. The obtained material was cooled and crystallized for 20 h at RT, filtered and washed on the filter with distilled water at neutral pH. Finally, the material was dried at 80 °C at constant mass.

### 2.2. Fly-Ash@Fe$_3$O$_4$ Adsorbent Synthesis

The hybrid compound was prepared by mixing the NaOH-treated ash with a solution of iron salts, followed by precipitation under an alkaline environment. Thus, 2.85 g FeCl$_2$·4H$_2$O and 5.7 g FeCl$_3$·6H$_2$O were dissolved and mixed in 15 mL ultrapure water, followed by filtration (220 nm pores). Then, 10 g of ash was added and sonicated for 5 min. The mixture was added to 300 mL ultrapure water at about 100 °C, under magnetic stirring (1000 rpm). Then, 15 g of NaOH was progressively added to the solution/suspension. After 5 min, the heating was stopped while the mixing was allowed to continue for an hour. Finally, the hybrid magnetic particles were washed with ultrapure water until the pH became 6.5–7. The obtained compound (about 9.3 g) was dispersed in 100 mL ultrapure water and kept at RT for characterization and adsorption experiments.

### 2.3. Characterization

The morphology and chemical composition of the magnetic material was established using a JEOL JSM-6390 equipped with an EDS detector (Jeol USA Inc., Brno-Kohoutovice, Czech Republic). The X-ray diffraction patterns were measured using a Brucker AXS D8-Advance powder X-ray diffractometer, CuK radiation = 0.1541 nm (Bruker, Brno, Czech Republic). A vibrating sample magnetometer, the Lake Shore 7410 (Lake Shore Cryotronics, Inc., Westerville, OH, USA), was used to obtain the magnetization data. A Microtrac/Nanotrac252 instrument (Microtrac, Montgomeryville and York, Pennsylvania, USA) was used to determine the colloidal nanoparticles' mean hydrodynamic diameter and size distribution. FTIR analysis was performed with a Jasco FT/IR-6100 spectrophotometer (JASCO Deutschland GmbH, Pfungstadt, Germany).

### 2.4. Congo Red Dye Characteristics and Preparation

Congo red is an anionic diazo dye with four separate color-assisted functional groups [40]. Congo red is the sodium salt of benzidinediazo-bis-1-naphthylamine-4-sulfonic acid, a diazo dye that is employed primarily as an indicator and a biological stain that turns red in alkaline solution and blue in acid solution [41]. Sigma Aldrich supplied the Congo red dye, which was used to prepare a 0.5 g/L stock solution.

### 2.5. Batch Adsorption Experiments

The effect of working parameters, which included contact time, adsorbent dosage, and initial concentration was analyzed (Table 1). All studies were conducted in a natural pH environment, at ambient temperature, with intermittent stirring. An external magnetic field was used to separate the synthesized magnetic adsorbent from solution (using a NdFeB permanent magnet). A UV-vis spectrophotometer set to 497 nm was used to detect the Congo red dye content in the supernatant solution (all experiments were realized in duplicate). Equations (1) and (2) were used to calculate the adsorption capacity, $q_e$ (mg/g), at equilibrium and at different time intervals, $q_t$ (mg/g):

$$q_e = \frac{(C_0 - C_e)V}{w} \tag{1}$$

$$q_t = \frac{(C_0 - C_t)V}{w} \tag{2}$$

where: $C_0$, $C_e$ and $C_t$ are the initial concentration, equilibrium concentration, and concentration at different time intervals (mg/L), respectively; $q$ is the quantity of Congo red dye adsorbed (mg/g); $V$ is the volume of Congo red dye solution (L); $w$ is the quantity of magnetic adsorbent used (g).

**Table 1.** Summary of applied experimental conditions in adsorption process of Congo red dye.

| Adsorbent Dosage, mg | Initial Congo Red Dye Concentration, mg/L | Contact Time, min |
|---|---|---|
| 3.5–20 | 10–100 | 5–120 |
| conc: 30 mg/L, contact time: 24 h | adsorbent dosage: 10 mg, contact time: 24 h | adsorbent dosage: 10 mg, conc: 30 mg/L |

*2.6. Isotherm and Kinetics Adsorption Study*

The experimental results were fitted by: (i) two isotherm models—Langmuir and Freundlich (in their linear and nonlinear forms); (ii) three kinetic models— the pseudo-first-order, pseudo-second-order, and intraparticle diffusion models. The equations used are presented in the Table 2. OriginPro 8.6E software was used to fit the nonlinear equations.

**Table 2.** Equations used for isotherm and kinetics adsorption study.

| | |
|---|---|
| Langmuir | Linear form : $\frac{C_e}{q_e} = \frac{1}{K_L q_{max}} + \frac{C_e}{q_{max}}$<br>Non $-$ linear form : $q_e = \frac{q_{max} K_L C_e}{1 + K_L C_e}$ |
| Freundlich | Linear form : $\ln q_e = \left(\frac{1}{n}\right)\ln C_e + \ln K_F$<br>Non $-$ linear form : $q_e = K_F C_e^{1/n}$ |
| Pseudo-first-order model | $\log(q_e - q_t) = \log q_e - \frac{(k_1 t)}{2.303}$ |
| Pseudo-second-order model | $\frac{t}{q_t} = \frac{1}{k_2 q_e^2} + \frac{t}{q_e}$ |
| Intraparticle diffusion model | $q_t = K_{ID} t^{0.5} + C$ |

where: $q_{max}$ is the maximum adsorption capacity (mg/g); $K_L$ is the Langmuir constant (L/g); $K_F$ is the Freundlich constant; $1/n$ (dimensionless) is a constant indicative of adsorption intensity; $q_t$ and $q_e$ (mg/g) represent the amount of dye adsorbed at time $t$ and at equilibrium, respectively; $k_1$ (1/min) and $k_2$ (g/mg min) are the pseudo-first-order rate and pseudo-second-order rate constants; $K_{ID}$ and C the intraparticle diffusion rate constant and another kinetic constant, respectively.

## 3. Results and Discussion

*3.1. Characterization of Magnetic Adsorbent*

The characterization of the magnetic adsorbent is presented in Figure 1.

The morphological structure of the unmodified fly ash and the magnetic adsorbent is shown in Figure 1a,b. In previous studies, it was demonstrated that the unmodified fly ash particles typically have a smooth surface texture and a predominantly spherical shape [37]. In comparison to unmodified fly ash, the magnetic material exhibits noticeable surface changes after treatment with NaOH and $Fe_3O_4$. The images show that both reagents used in the synthesis have been distributed uniformly on the surface of the unmodified fly ash. Magnetic adsorbent crystallites with various shapes can also be observed on the surface.

The elemental composition (Figure 1c) shows that the magnetic adsorbent contains the same components as unmodified fly ash, i.e., O, Na, Mg, Al, Si, K, Ca, Ti, and Fe. However, the results obtained show that the treatment of fly ash with NaOH (in the first stage), followed by treatment with $Fe_3O_4$ (the second stage of the synthesis), changed its chemical composition, particularly with regard to Fe content. For the unmodified fly ash, the Fe content is 0.94% [37], while for the prepared material the Fe content is 18.77%.

Figure 1d shows the diffractograms of the unmodified fly ash, the prepared magnetic adsorbent and $Fe_3O_4$. The XRD analysis demonstrated the crystalline phases coming from

the unmodified fly ash, such as mullite (16.61°, 33.21°, 40.71) and quartz (20.71°, 26.71°, 50.32°). The XRD diffractogram of $Fe_3O_4$ shows characteristics peaks at 30.29°, 35.27°, 43.15°, 53.74°, 57°, and 62.82°. These peaks can also be observed in the prepared magnetic adsorbent. Because the patterns provided by the XRD for magnetite and maghemite are similar, one cannot discriminate between them. However, we can assume that a combination of the two types of materials formed core–shell particles, with the core based on magnetite and the shell made up of its oxidized form, i.e., maghemite. Overall, the diffraction peaks indicated that the synthesis of the Fly-ash@$Fe_3O_4$ mixture was successfully performed.

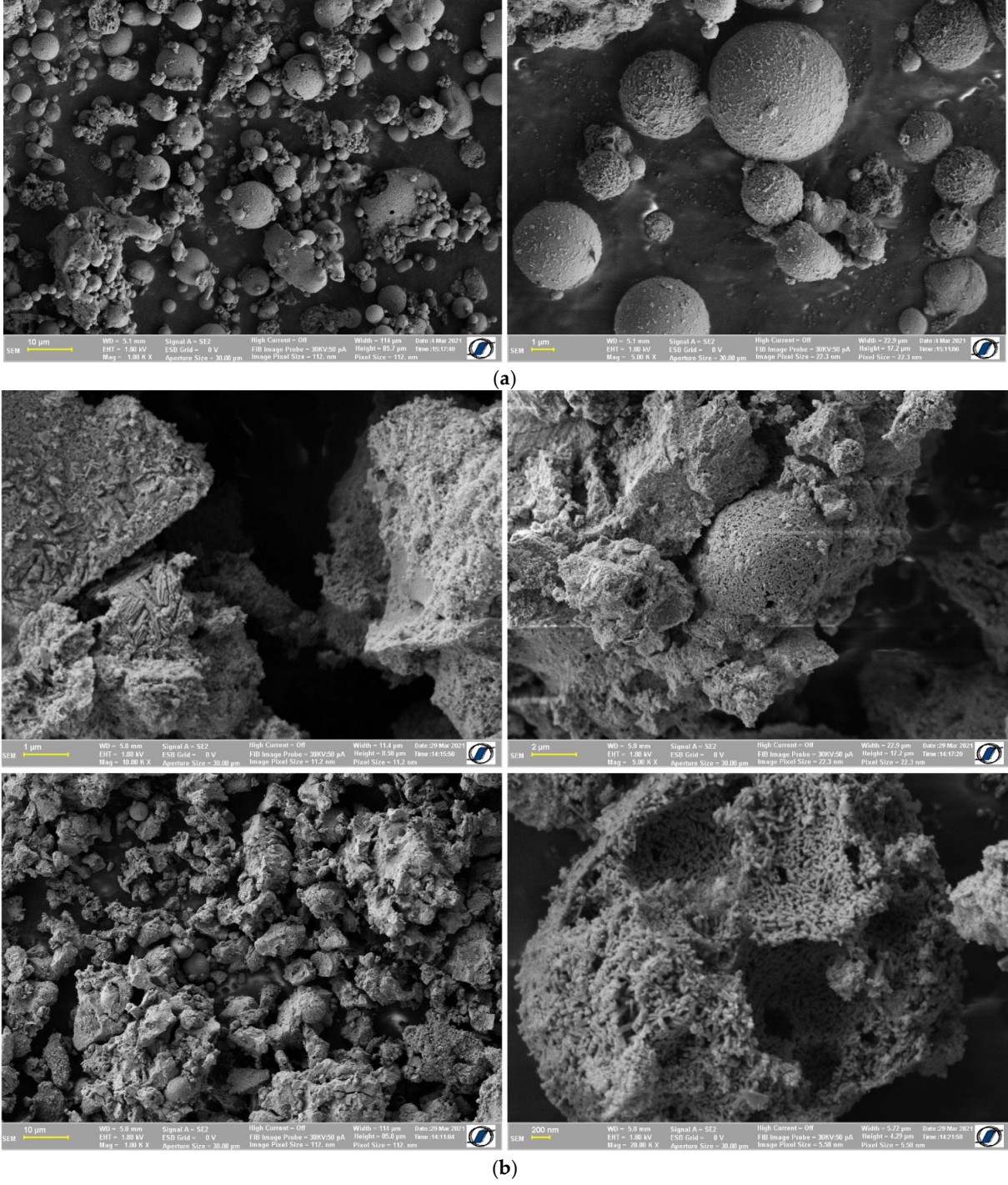

**Figure 1.** *Cont.*

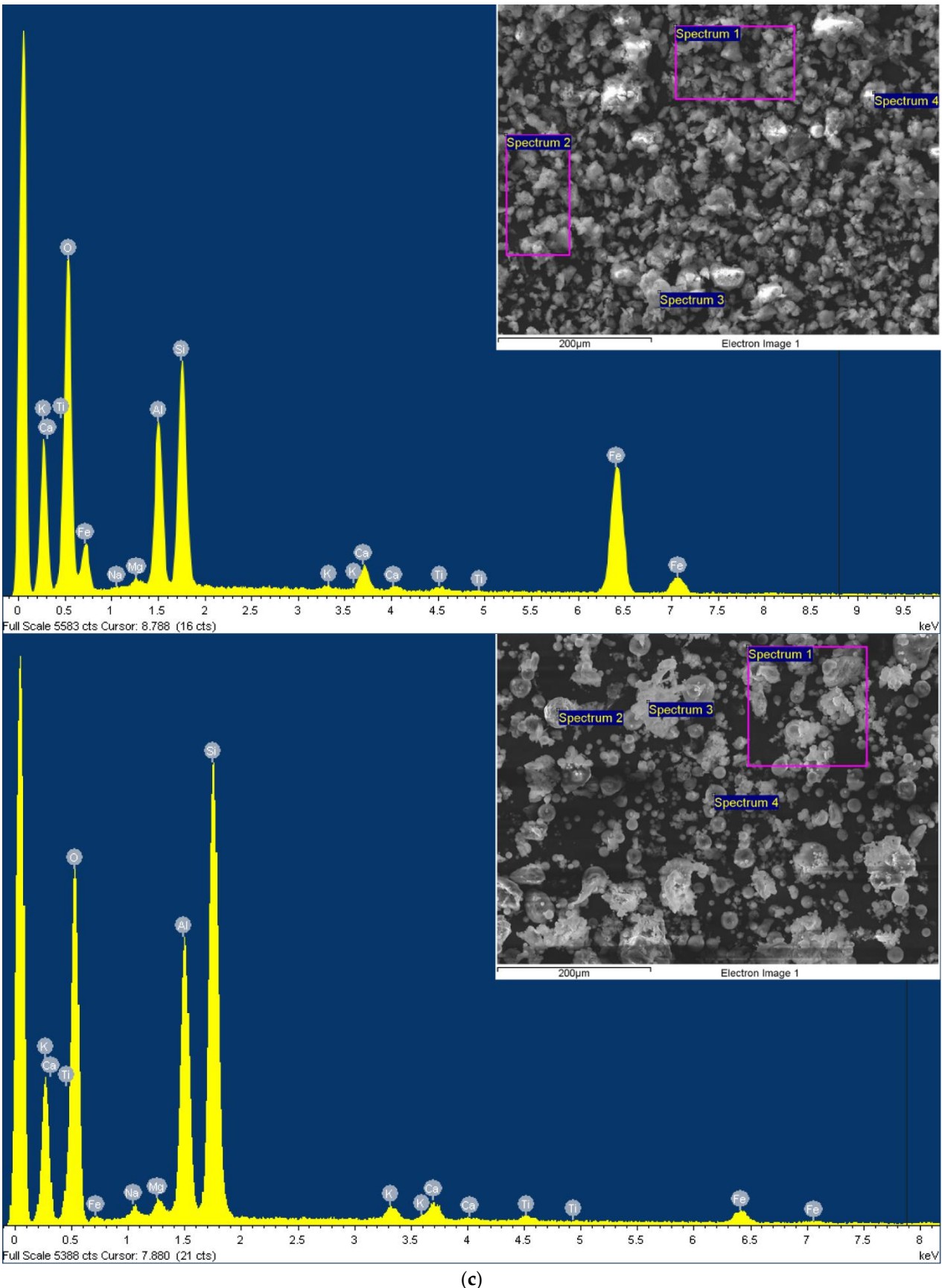

(**c**)

**Figure 1.** *Cont.*

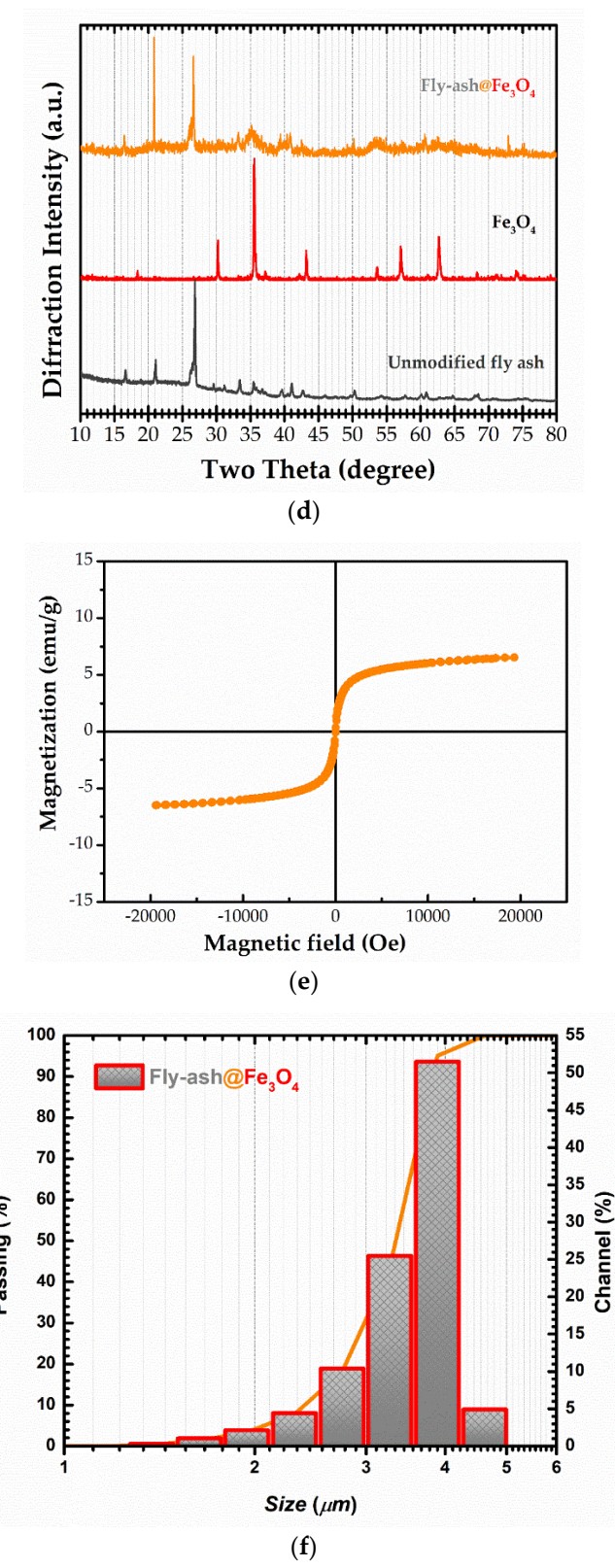

(**d**)

(**e**)

(**f**)

**Figure 1.** Characterization of unmodified fly ash and Fly-ash@Fe$_3$O$_4$ materials:(**a**) SEM images of unmodified fly ash at different magnifications; (**b**) SEM images of Fly-ash@Fe$_3$O$_4$ material at different magnifications; (**c**) EDS spectrum of unmodified fly ash and Fly-ash@Fe$_3$O$_4$; (**d**) XRD analysis of unmodified fly ash and Fly-ash@Fe$_3$O$_4$; (**e**) VSM analysis of Fly-ash@Fe$_3$O$_4$; (**f**) DLS analysis of Fly-ash@Fe$_3$O$_4$.

The field dependence of the magnetization of the hybrid compound is presented in Figure 1d. The hysteresis loop displays a value of saturation magnetization of 6.51 emu/g at an applied field of 20 kOe and temperature of 300 K. The field coercivity, Hc, of 21 Oe and remanent magnetization, Mr, of 0.23 emu/g indicate superparamagnetic behavior.

The mean diameter of the hybrid particles obtained by DLS was about 3.34 μm, with the distribution of sizes ranging from 0.31 μm to 4.89 μm (Figure 1f). Since the dimensions obtained via DLS takes into account the hydrodynamic diameter, the analyzed particles often seem to be bigger [42]. The suspension was unstable over time, with settled fraction visible from the first day. Taking into account the size and density of the particles, this was an expected phenomenon.

### 3.2. Effect of Working Parameters on the Adsorption Process

The results related to the effect of the working parameters, i.e., magnetic adsorbent dosage, initial Congo red dye concentration, and contact time are presented in Figure 2. In order to establish how the magnetic adsorbent dosage influences the adsorption capacity, different quantities of material were mixed with Congo red dye solution, using an initial concentration of 30 mg/L at RT. Using an adsorbent dosage of 10 mg/10 mL solution at RT for a contact time of 24 h, the effect of the initial Congo red dye concentration (10 mg/L, 20 mg/L, 30 mg/L, 40, mg/L, 50 mg/L, 60 mg/L, 80, mg/L, and 100 mg/L) was examined. The adsorption kinetic tests were carried out for different time intervals using initial Congo red dye concentration of 30 mg/L.

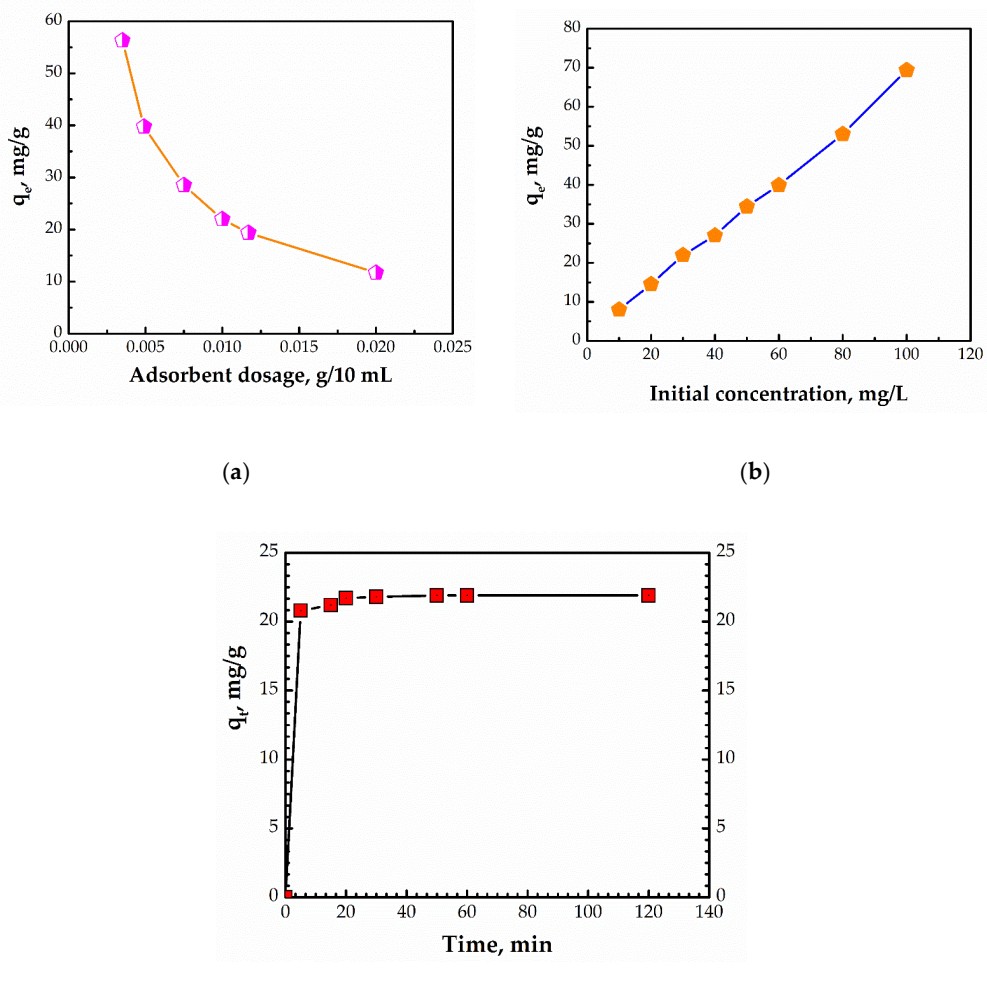

**Figure 2.** Effect of (**a**) magnetic adsorbent dosage; (**b**) initial Congo red dye concentration; (**c**) contact time.

Further, the influence of increasing the dosage of magnetic adsorbent from 3.5 mg to 20 mg, for a concentration of 30 mg/L and a volume of 10 mL, on the adsorption capacity of the Congo red dye was investigated. An adsorption capacity of 56.3 mg/g was obtained using a dosage of 3.5 mg/10 mL Congo red solution. A considerable reduction in the adsorption capacity was noticed as the adsorbent dosage increased.

Figure 2b illustrates the effect of the initial Congo red dye concentration on the adsorption capacity of the magnetic material. According to the obtained results, it can be stated that this parameter has a notable impact on the adsorption capacity. The lowest values were obtained at low dye concentrations, while values higher than 50 mg/g were obtained at initial dye concentrations of 80 mg/g and 100 mg/g.

Moreover, it was found that the adsorption capacity was significantly impacted by the contact time. The adsorption capacity increased with the increasing contact time, as follows: from 20.8 mg/g to 21.7 mg/g in the first 20 min, after which the adsorption capacity remained almost unchanged. The Congo red and the material were successfully attracted by electrostatic forces due to the availability of large and empty adsorption sites. A similar trend was also noted by Harja and collaborators using unmodified fly ash [34]. On the other hand, Ahsani-Namin and co-workers demonstrated that "the trend of the adsorption capacity vs. contact time of meso/microporous CuO/ZnO nanostructures has two stages: a rapid increase in the initial times and a slowly rising in final times" [43]. It should be noted that the initial Congo red dye concentration used in both of the abovementioned studies was 30 mg/L. In conclusion, the data obtained in this study reveal that a contact time of about 120 min is adequate to attain the equilibrium, as shown by the adsorption findings presented in Figure 2c.

Moreover, for the unmodified fly ash and Fly-ash/NaOH materials, a study regarding the effect of contact time was performed in order to emphasize the purpose of this study, that is, the improvement of the adsorption capacity by treating the material with $Fe_3O_4$. Figure 3 illustrates the patterns for the adsorption of Congo red dye on the three materials, and it is evident that the adsorption capacity is greater for the magnetic adsorbent than that of unmodified fly ash and Fly-ash/NaOH materials for all the time intervals investigated. The adsorption capacity of the magnetic adsorbent is about 1.92 higher compared with unmodified fly ash, and about 1.38 compared with Fly-ash/NaOH after 120 min of contact time. The enhanced adsorption capacity of the magnetic adsorbent can be attributed to the effect of both new products formed by the NaOH attack and $Fe_3O_4$ on the unmodified fly ash surface.

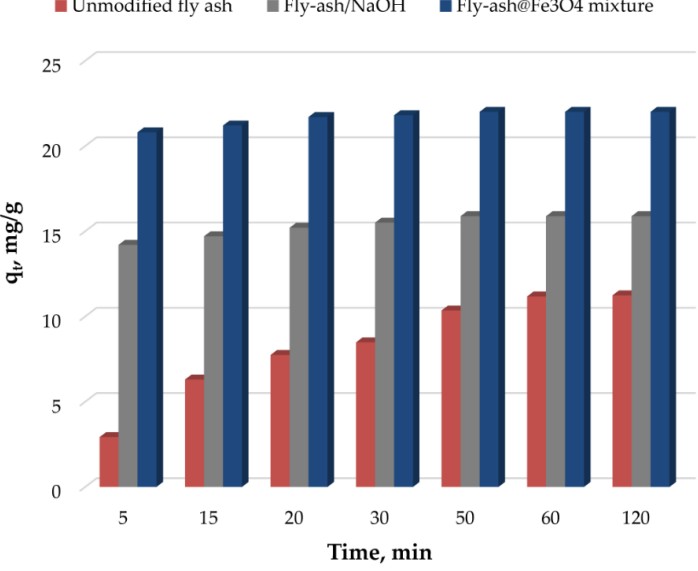

**Figure 3.** Comparison of unmodified fly ash, Fly-ash/NaOH, and Fly-ash@$Fe_3O_4$ adsorbents.

The experimental data (Figure 3) demonstrated that in the case of the magnetic adsorbent, the time required to reach the maximum adsorption capacity was reduced (15–20 min), which has positive effects on the energy consumption required for agitation.

Figure 4 depicts the fitting curves of adsorption isotherms, and kinetic models while Tables 3 and 4 depict the adsorption isotherm constants and kinetic parameters' values, respectively.

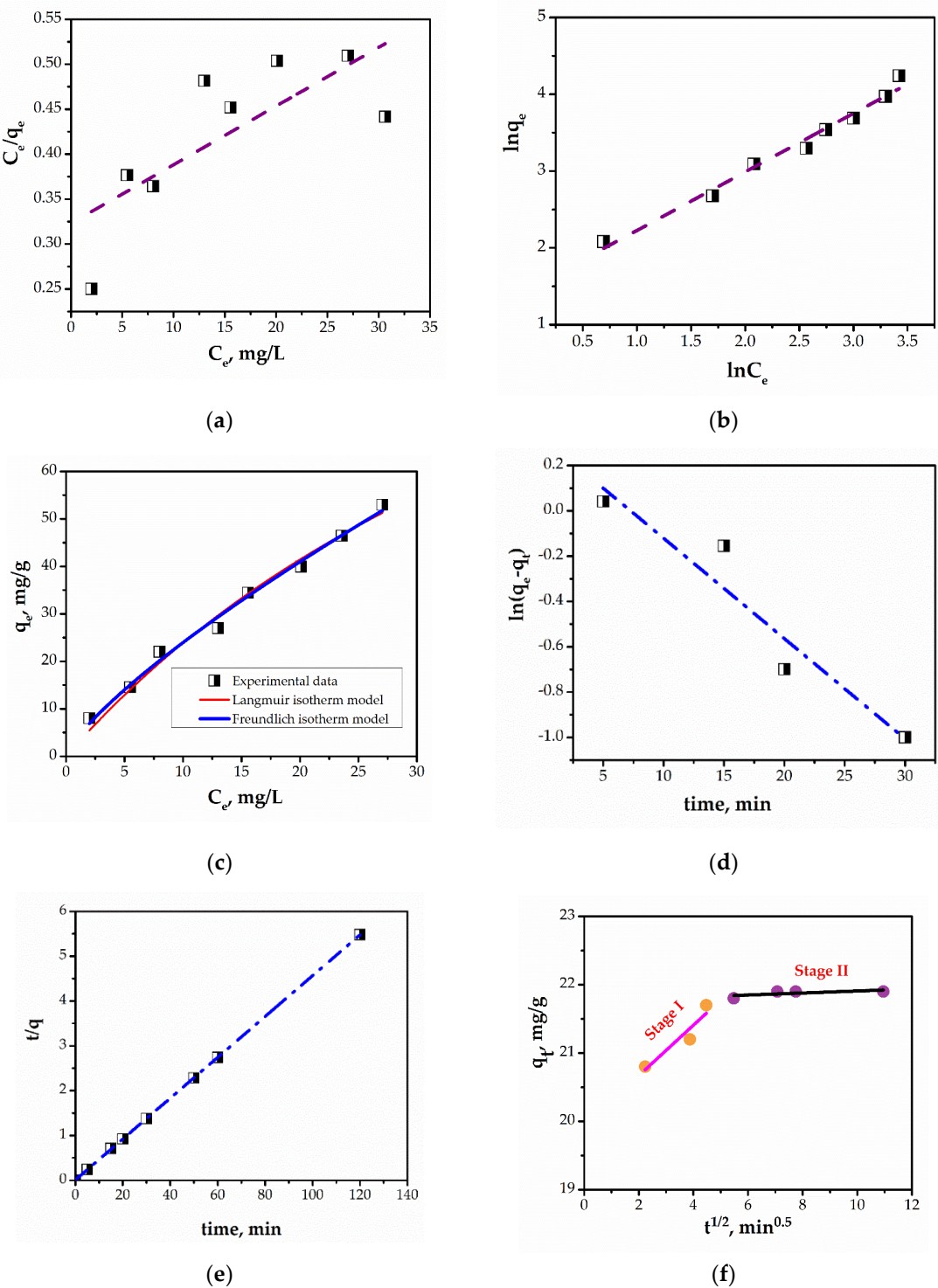

**Figure 4.** Fitting curves of (**a**) Langmuir model; (**b**) Freundlich model; (**c**) isotherm models in nonlinear form; (**d**) pseudo-first-order model; (**e**) pseudo-second-order model; (**f**) intraparticle diffusion model.

**Table 3.** Adsorption isotherm constants.

| Linear Models | | Non-Linear Models | |
|---|---|---|---|
| Langmuir | Freundlich | Langmuir | Freundlich |
| $q_{max}$ = 153.85<br>$K_L$ = 0.0201<br>$R^2$ = 0.5716 | $K_F$ = 4.33<br>$1/n$ = 0.7617<br>$R^2$ = 0.9816 | $q_{max}$ = 157.21<br>$K_L$ = 0.0179<br>$R^2$ = 0.9829 | $K_F$ = 4.028<br>$1/n$ = 0.774<br>$R^2$ = 0.9905 |

**Table 4.** Kinetic parameters.

| Experimental $q_e$, mg/g | Pseudo-First Order Model | Pseudo-Second Order Model | Intraparticle Diffusion Stage I | Intraparticle Diffusion Stage II |
|---|---|---|---|---|
| 21.9 | $k_1$ = 0.1018<br><br>$R^2$ = 0.9178 | $q_e$ cal = 21.93<br>$k_2$ = 0.2079<br>$R^2$ = 1 | $K_{ID}$ = 0.3691<br>C = 19.932<br>$R^2$ = 0.8975 | $K_{ID}$ = 0.0147<br>C = 21.76<br>$R^2$ = 0.4578 |

Langmuir and Freundlich isotherms are often applied, regardless of the investigated pollutant, to obtain physical–chemical information about the adsorption process [44–46]. The Langmuir model depicts the distribution of pollutant between the solid and liquid phases whereas the Freundlich model is used in order to establish the adsorption process on the heterogeneous surface of the adsorbent, [47].

In the linear form, the maximum adsorption capacity, $q_{max}$, and Langmuir constant, $K_L$, are 153.85 mg/g and 0.0201 L/g, respectively. However, in its nonlinear form, as shown in Table 3, the maximum adsorption capacity is 157.21 mg/g and $K_L$ is 0.0179 L/g. When considering the correlation coefficients ($R^2$), the fitting findings demonstrated that the Freundlich model provided a higher correlation than the Langmuir model in both the linear and nonlinear forms. The value of ''n'' indicates a favorable and physical process. Consequently, it can be stated that the investigated pollutant can be absorbed by multilayer adsorption.

According to the pseudo-first-order kinetic model, the rate of adsorption is directly proportional to the difference between the concentration of the adsorbate at different times "t" and the initial concentration. The adsorption is attributed to physicochemical interactions between the adsorbate and adsorbent phases in the pseudo-second-order kinetic model, which assumes that chemisorption is the rate-regulating step. According to the intraparticle diffusion model, the rate of metal adsorption is influenced by how quickly the pollutant diffuses onto the adsorbent surface [37]. By analyzing the data shown in Table 4, it can be observed that the correlation coefficient, $R^2$, for the pseudo-first-order kinetic model is below 0.92, and the reaction rate constant, $k_1$, is 0.1018 1/min. The $k_2$ constant, corresponding to the pseudo-second-order kinetic model, and the $R^2$ are 0.2079 g/mg·min and 1, respectively. On the other hand, according to the intraparticle diffusion model, the adsorption process of Congo red dye on magnetic adsorbent occurs in two stages: the first stage corresponds to a time interval between 5–20 min with an $R^2$ value of 0.8975, and the second one corresponds to a time interval between 30–120 min with $R^2$ values below 0.5. Based on the correlation coefficient, $R^2$, it can be stated that Congo red dye adsorption on magnetic adsorbent is in accordance with the pseudo-second-order kinetic model compared with the pseudo-first-order model and intraparticle diffusion model, respectively. The similar $q_e$ exp and $q_e$ cal (21.93 mg/g vs. 21.9 mg/g) values also supported the suitability of the pseudo-second-order model to explain the adsorption kinetic data.

The findings obtained through the kinetic study (pseudo-second-order kinetic model fits the adsorption data) revealed a chemisorption process that involves an electrostatic attraction between the Fly-ash@$Fe_3O_4$ adsorbent and the Congo red dye. The representation of the proposed adsorption mechanism of Congo red dye by Fly-ash@$Fe_3O_4$ is depicted in Figure 5.

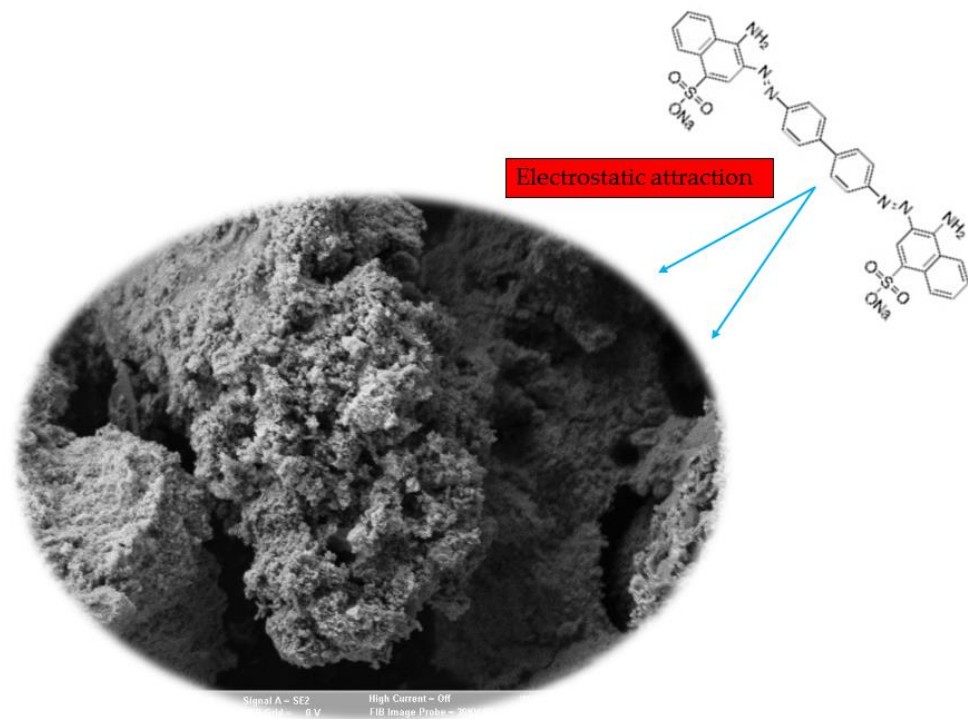

**Figure 5.** The proposed adsorption mechanism of Congo red dye by Fly-ash@Fe$_3$O$_4$.

### 3.3. Comparison with the Available Literature and Circular Economy Concept

Table 5 shows a comparison of the maximum adsorption capacities of different magnetic adsorbents for Congo red dye removal from aqueous solution presented in the literature.

**Table 5.** Comparison of maximum adsorption capacities of different magnetic adsorbents for Congo red.

| Adsorbent | $q_{max}$, mg/g | References |
|---|---|---|
| FANiFe$_{50}$ | 22.73 | [48] |
| $\gamma$-Fe$_2$O$_3$-sepolite-NH$_2$ | 126.4 | [49] |
| ZnFe$_2$O$_4$/SiO$_2$/Tragacanth gum magnetic nanocomposite | 128.2 | [50] |
| Fe$_3$O$_4$@SiO$_2$ | 14.76 | [51] |
| Fe$_3$O$_4$@SiO$_2$@Zn$-$TDPAT | 17.73 | [51] |
| Cellulose acetate/chitosan/SWCNT/Fe$_3$O$_4$/TiO$_2$ composite nanofibers | 74.2 | [52] |
| 3D flower-like maghemite particles | 102.7 | [53] |
| Magnetic peanut husk | 56.3–79 | [54] |
| Magnetite-nanoparticle-decorated NiFe layered double hydroxide | 79.6 | [55] |
| Fe doped ZnO nano particles | 93.75 | [56] |
| NH$_2$-Fe$_3$O$_4$-GO-MnO$_2$-NH$_2$ | 54.95 | [57] |
| Polycrystalline $\alpha$-Fe$_2$O$_3$ nanoparticles | 58.2 | [58] |
| Iron oxide/activated carbon (Fe$_3$O$_4$/AC) nanocomposite | 122.22 | [59] |
| Graphene oxide-CuFe$_2$O$_4$ nanohybrid | 114.21 | [60] |
| Fe$_3$O$_4$@10%Zn | 59 | [61] |
| **Fly-ash@Fe$_3$O$_4$** | **153** | **This work** |

In order to demonstrate the circular economy concept, a reusability study was performed with three cycles of adsorption/desorption using 10 mg of magnetic adsorbent per 10 mL Congo red dye solution, at neutral pH for 60 min of contact time. The desorption agent was 0.1 M of NaOH. The results demonstrated that the adsorption capacity of the magnetic adsorbent decreased slightly after three adsorption/desorption cycles (from 21.9 mg/g to 20.5 mg/g). Consequently, the findings of the present investigation suggest that the circular economy concept may be used to create an effective magnetic adsorbent based on the treatment of an industrial and abundant waste with NaOH, 10 M, and $Fe_3O_4$, without using additional chemicals or energy.

### 3.4. Characterization of Fly-Ash@$Fe_3O_4$ Loaded Adsorbent

XRD and FTIR techniques were also performed after Congo red dye adsorption (Figure 6).

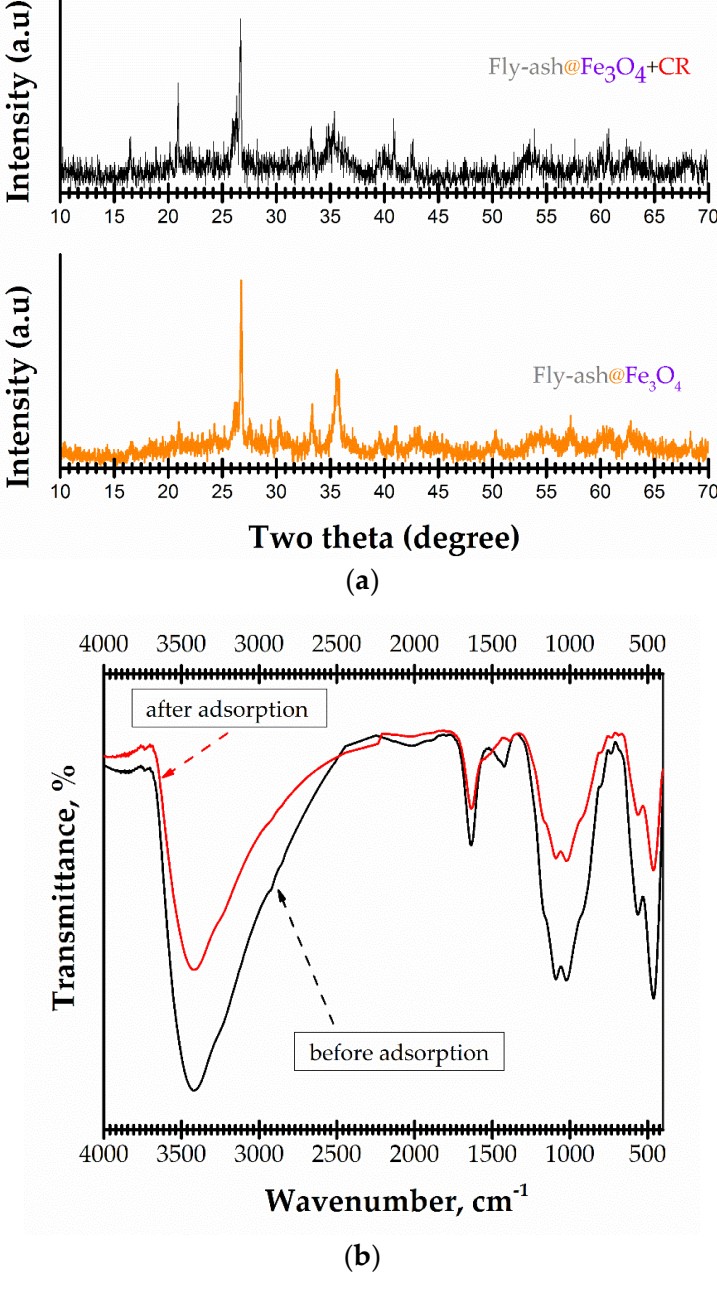

**Figure 6.** Characterization of Fly-ash@$Fe_3O_4$-loaded adsorbent: (**a**) XRD analysis; (**b**) FTIR analysis.

A comparison of the XRD and FTIR data before and after the Congo red dye adsorption process using Fly-ash@$Fe_3O_4$ material (Figure 6a,b) showed that there were some modifications, including a slight shift in the peaks and different peaks' intensities, which may be a marker of Congo red dye adsorption.

## 4. Conclusions

In the present work, a material based on the treatment of fly ash with NaOH and $Fe_3O_4$ is proposed for use in water contaminated with Congo red dye. The main advantage of the suggested approach is based on the fact that this method is easy, low-cost and highly efficient Moreover, this type of mixture represents an innovative category of magnetic materials with excellent adsorption capacities.

The physical characteristics of the prepared magnetic adsorbent demonstrated that the material was successfully synthesized. Even if the saturation magnetization is rather low, the magnetic adsorbent can be easily separated with an external magnet field.

The dose of the magnetic adsorbent, the contact period, and the initial concentration were investigated as working parameters. The mixture of unmodified fly ash treated with NaOH and $Fe_3O_4$ leads to a higher adsorption capacity compared to unmodified fly ash or NaOH-treated fly ash, as indicated by our investigation of the contact time: 11.24 mg/g and 15.88 mg/g for unmodified fly ash and Fly-ash/NaOH adsorbents, respectively, vs. 22 mg/g for the magnetic adsorbent.

The data were fitted using the Langmuir (its linear and nonlinear forms) and Freundlich isotherm and three kinetic models (pseudo-first-order kinetic model, pseudo-second-order kinetic model, and intraparticle diffusion model). According to the findings of the equilibrium isotherms and kinetic study, the adsorption of Congo red dye matched the Freundlich isotherm and pseudo-second-order model, respectively. The prepared magnetic adsorbent displays a maximum adsorption capacity of 153 mg/g and a fast adsorption rate (to attain the equilibrium, there was a contact time of approximately 20 min).

The circular economy idea was supported by the reusability study. The XRD and FTIR results obtained after the adsorption process confirmed that the Congo red dye molecules were successfully attached on the surface of Fly-ash@$Fe_3O_4$ material.

Overall, all the data obtained in this study support the synthesis of this type of magnetic adsorbent based on the mixture of unmodified fly ash, 10 M NaOH solution and $Fe_3O_4$, with applicability for the treatment of wastewater contaminated with Congo red dye. Thus, the problems caused by the presence of this type of pollutant can be reduced.

**Author Contributions:** Conceptualization, G.B. and M.H.; methodology, G.B. and N.L.; formal analysis, G.B. and D.-D.H.; investigation, G.B., D.-D.H. and M.H.; writing—original draft preparation, G.B., D.-D.H. and M.H.; writing—review and editing, M.H., N.L. and H.C.; visualization, H.C.; supervision, H.C.; project administration, N.L.; funding acquisition, N.L. All authors have read and agreed to the published version of the manuscript.

**Funding:** This work was supported by a grant of the Ministry of Research, Innovation and Digitization under the Nucleu programme, Project PN 19 28 01 01 and POC/163/1/3/Innovative Technology Project, project number 399/390075/17.11.2021, Cod SMIS 2014: 120951 (synthesis of Fly-ash/NaOH material).

**Institutional Review Board Statement:** Not applicable.

**Informed Consent Statement:** Not applicable.

**Data Availability Statement:** The data presented in this study are available on request from the corresponding author.

**Conflicts of Interest:** The authors declare no conflict of interest.

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
