# Peer review of "Studies on the Removal of Congo Red Dye by an Adsorbent Based on Fly-Ash@Fe3O4 Mixture"

_magnetochemistry, doi:10.3390/magnetochemistry8100125_

Round 1

Reviewer 1 Report

This manuscript describes the preparation and use of magnetite loaded fly ash composite for the adsorption of Congo red. The produced composite is characterized using SEM, EDS, XRD, VSM, and DLS. The sorption was investigated through Batch method. The Isotherm and kinetics adsorption were studied, but the experiments were performed one time without respect the reproducibility. In this regard, I would like to recommend it to be published after major revision.

Comments

1-        The authors should repeat the experiments two or three times at least to show the reproducibility of adsorbent.

2-        Figure 1c and Figure 1f- The quality of these spectrum are poor and should be improved, also the authors should provide that eds spectrum of the fly ash before treatment to show the diffrence.

3-        Because the XRD patterns of magnetite and maghemite are very similar, the authors cannot state unequivocally that the magnetic phase is magnetite. The authors should add a comment about this

Author Response

Dear Reviewer 1,

We want to thank you for taking the time to review this article and for your constructive suggestions for improving the scientific quality of our manuscript. All modifications were clearly marked and highlighted in yellow color in the revised manuscript.

This manuscript describes the preparation and use of magnetite loaded fly ash composite for the adsorption of Congo red. The produced composite is characterized using SEM, EDS, XRD, VSM, and DLS. The sorption was investigated through Batch method. The Isotherm and kinetics adsorption were studied, but the experiments were performed one time without respect the reproducibility. In this regard, I would like to recommend it to be published after major revision.

Comments

  • The authors should repeat the experiments two or three times at least to show the reproducibility of adsorbent.

Answer: Thanks for your suggestions, we repeated the experiments and tested the reproducibility. In manuscript it was completed with: ‘’all experiments were realized in duplicate’’, row 146.

  • Figure 1c and Figure 1f- The quality of these spectrum are poor and should be improved, also the authors should provide that eds spectrum of the fly ash before treatment to show the diffrence.

Answer: Thanks for your fine observation. We did all of our best to improve the quality of these figures. The resolution is 300 dpi, RGB color mode. Also, the EDS spectrum of unmodified fly ash was added in the new version of the manuscript.

3-        Because the XRD patterns of magnetite and maghemite are very similar, the authors cannot state unequivocally that the magnetic phase is magnetite. The authors should add a comment about this.

Answer: Thank you for this valuable observation. Accordingly, we have added in the manuscript the following sentence:

„Due to the similar patterns provided by XRD for magnetite and maghemite, one cannot discriminate between them. However, we can assume a combination between the two types of materials to form core-shell particles, with the core based on magnetite and the shell made up of its oxidized form, i.e. maghemite.”

Dr. Gabriela Buema

Reviewer 2 Report

Reviewer #2: This paper was assessed the effectiveness of the Fe3O4-loaded fly ash composite for the adsorption of Congo red dye. A material based on the treatment of fly ash with NaOH and Fe3O4 is proposed for waters contaminated with Congo red dye. The main advantage of this method is that it is a simple, low-cost and efficient method. Minor revision is suggested, and the following points are provided for the authors:

1.    Where are the advantages of innovation? It's just that no one has done it, and it's not enough to be an advantage as an innovation point.

2.    Fig. 1c cannot show the changes before and after the sample. It is recommended to supplement the SEM of the unmodified fly ash, and supplement the element distribution map of the adsorbent, and explain the formation mechanism.

3.    The hybrid particles obtained by DLS in this paper have an average diameter of about 3.34 µm, and a size distribution ranging from 0.31 µm to 4.89 µm (Fig. 1f). However, from the block shape of the sample seen in Figure 1ab, it is not seen that it forms an independent shape. Please supplement the SEM images of the sample at different magnifications so that the morphology of the sample can be seen more clearly.

4.    If you want to prove the perfect combination of Fe3O4 and fly ash, it should be compared with the XRD of unbound fly ash, and it is recommended to put it in the same figure for comparison.

5.    The XRD and FT-IR of the as-prepared sample and spent sample should be compared to confirm the stability of the sample, and explain the adsorption mechanism.

6.    The manuscript states that the properties of the prepared samples are much better than those of the unmodified fly ash, such as crystallinity, BET, zeta potential, etc. These points should be explained clearly.

7.    The adsorption experiments were performed at different pH values. The isoelectric point (IEP) is an important parameter often used to characterize the adsorption properties of materials. More discussion on the effect of isoelectric point (IEP) and pH on adsorption performance of the prepared samples should be added and the related works can be read and referenced.

Round 2

Reviewer 1 Report

The paper can accept in the recent form.

Thanks for your efforts

Reviewer 2 Report

The revised manuscript has been well modified, according to the comments. It is suggested to accept in the current form.